# Flow angle measurement of a yawed turbine and comparison to models

Tyler Gallant<sup>1</sup> and David A. Johnson<sup>1</sup>

<sup>1</sup>Wind Energy Group, Mechanical and Mechatronics Engineering, University of Waterloo, 200 University Ave W, Waterloo, Canada N2L 3G1

Correspondence to: David A. Johnson (David.Johnson@uwaterloo.ca)

**Abstract.** The torque generated by a wind turbine blade is dependent on several parameters, one of which is the angle of attack. Several models for predicting the angle of attack in yawed conditions have been proposed in the literature, but there is a lack of experimental data to use for direct validation.

- To address this problem, experiments were conducted under controlled conditions at the University of Waterloo Wind Gen-5 eration Research Facility using a 3.4 m diameter test turbine. A five-hole pressure probe was installed in a modular 3D printed blade and was used to measure the angle of attack,  $\alpha$ , as a function of several parameters. Local flow angle measurements for all azimuthal angles were obtained at radial positions of r/R = 0.55 and 0.72 at tip speed ratios ( $\lambda$ ) of 5.0, 3.6, and 3.1. The yaw offset of the turbine was varied from -15° to +15°. Span-wise flow angle measurements are presented for the r/R = 0.55cases, and show the variation in radial flow direction throughout yawed rotation.
- Experimental results were compared directly to angle of attack values calculated using a model proposed by Morote in 2015. Modeled values were found to be in close agreement with the experimental results. The angle of attack was shown to vary cyclically in the yawed case while remaining mostly constant when aligned with the flow, as expected.

These five-hole probe measurements were also used to characterize the upstream flow profile. Wind speeds determined using the five-hole probe measurements are presented and are in agreement with measurements obtained in the wind facility during testing. The quality of results indicates the potential of the developed instrument for wind turbine measurements.

# 1 Introduction

15

As developments in wind turbine design continue, so too does the need for accurate experimentation and testing of new wind turbine blade designs. One variable that has a critical impact on the performance of a wind turbine blade is the angle of attack, which is directly related to the forces generated on the blade. Conventionally during wind turbine testing, the upstream wind

20 speed is measured via a separate meteorological tower upstream of the turbine, and this wind speed is used in combination with the wind turbine geometry and operational characteristics to model the angle of attack. However, upstream wind speed measurements do not accurately represent the wind speed at the turbine blades for many reasons including the axial induction of the rotor. The angle of attack is therefore calculated using models (*e.g.* the blade-element momentum method) using

5

measurements several steps removed from the leading edge of the blade. The result is a lack of accurate experimental data for validating theoretical models.

To address this problem, a five-hole pressure probe has been mounted to the leading edge of a rotating wind turbine blade to accurately measure the local flow angle (LFA) to determine the angle of attack,  $\alpha$ , in the rotational plane. Experiments were conducted to measure the variation in  $\alpha$  as a function of the azimuthal angle of the blade,  $\phi$ , the tip speed ratio  $\lambda$ , and the yaw angular position,  $\gamma$ . Experimental results were then compared to theoretical results calculated using a model proposed by Morote (2015). Span-wise flow angle measurements and the upstream wind profile were also measured.

# 2 Background

The use of five-hole pressure probes to measure the wind turbine blade angle of attack and span-wise flow angle is not a new concept, though the method is relatively uncommon in the literature. The most notable example of such studies are the NREL Unsteady Aerodynamics Experiments (UAE) (Hand et al., 2001). Phase VI of those experiments involved a 10 m diameter, stall regulated 20 kW wind turbine consisting of two twisted and tapered blades. The turbine was well equipped with measurement instrumentation, including five-hole pressure probes. While the Phase VI report (Hand et al., 2001) does not include experimental results from the project, the report does provide equations necessary to convert flow pitch and span-wise flow angles relative to the probe to flow angles with respect to the blade. The Local Flow Angle (LFA) and Span-wise Flow

Angle (SFA) relative to the blade chord are defined as (Hand et al., 2001):

$$LFA = \arctan \frac{\cos \alpha_p \cos \left(\beta_p + \epsilon\right) \sin \theta + \sin \alpha_p \cos \theta}{\cos \alpha_p \cos \left(\beta_p + \epsilon\right) \cos \theta - \sin \alpha_p \sin \theta} \tag{1}$$

$$SFA = \arctan \frac{\cos \alpha_p \sin \left(\beta_p + \epsilon\right)}{\cos \alpha_p \cos \beta_p + \epsilon \cos \theta - \sin \alpha_p \sin \theta} \tag{2}$$

20

where  $\alpha_p$  is the local flow angle with respect to the probe,  $\beta_p$  is the span-wise flow angle with respect to the probe,  $\epsilon$  is the span-wise probe angle offset and  $\theta$  is the local flow probe angle offset. These angles are shown in Figures 1 and 2 where the SFA is positive when flow moves outboard toward the tip. Local flow angle measurements from the NREL experiments (Hand et al., 2001) were analyzed and presented as angle of attack measurements in Schepers and van Rooij (2008) after a correction for induction due to the bound circulation was applied via the Biot-Savart law. Similar corrections were applied for other pressure probe experiments by Butterfield (1989) and surface pressure methods by Shen et al. (2009).

Five-hole pressure probes at 38.5% and 55.8% radius were used by Maeda and Kawabuchi (2005) to measure the angle of attack for a three-bladed, 10 m diameter, upwind turbine in the open environment with yaw positions ( $\gamma$ ) of -45° to +45° in increments of 15°. Results of their experiments showed that at a 0° yaw-offset the inflow velocity measured throughout the blade rotation (azimuthal angle  $\phi$ ) was essentially constant although they found a variation in  $\alpha$  of as much as 2.5° with azimuthal position that they described as the influence of the atmospheric boundary layer (Maeda and Kawabuchi, 2005).

Figure 1. Demonstrative diagram of flow angles relative to the airfoil profile

Figure 2. Demonstrative diagram of flow angles relative to the airfoil (plan view)

However, when the turbine was yawed  $\pm 45^{\circ}$  relative to the upstream wind direction, the inflow velocity fluctuated significantly with azimuthal position, increasing as the blade rotated towards the wind and decreasing as the blade rotated away from the wind. In the yawed conditions the angle of attack was observed to vary differently than the flow velocity, reaching a minimum when the velocity was at a maximum for the -45° yaw case and remaining relatively constant throughout the rotation during the +45° yaw case. The variation in the angle of attack was found to be as high as 10° throughout the blade rotation.

Directional pressure probes are not the only method being used to measure the angle of attack at a radial location on a wind turbine blade. In an early study Butterfield et al. (1991) used a flow angle probe to determine angle of attack at no yaw and a  $30^{\circ}$  yaw case showing significant variation in angle of attack near the tower of the downwind turbine. Johnson et al. (2012) conducted experiments using a three bladed 3.4 m diameter turbine installed in a large-scale wind generation

- research facility. Wind turbine blade performance was measured indirectly via simultaneous measurements of the wind turbine output and the wind velocity upstream and downstream of the rotor. Using the measured velocities, Johnson et al. (2012) used methods described by Hansen et al. (1998) and Johansen and Sørensen (2004) to calculate the axial and tangential induction factors, a and a', respectively. These values were then used to calculate the blade angle of attack. Johnson et al. (2012) found that the method resulted in angle of attack estimates within 15-20% of airfoil design data indicating that the use of velocity
- measurements to validate five-hole probe measurements may be appropriate.

Multi-hole probes have also been used to characterize the flow field upstream of a wind turbine. Recently, Petersen et al. (2015) developed a method for evaluating the upstream flow velocity, axial induction factor and inflow turbulence by extrapo-

lating flow measurements at the leading edge of the blade. The method begins with the actuator disc model, which relates the velocity at the wind turbine rotor,  $U_{disc}$ , to the upstream wind velocity,  $U_{\infty}$ , via the axial induction factor, *a*:

$$U_{disc} = U_{\infty}(1-a) \tag{3}$$

The axial velocity at the disc is calculated using the angle of attack and flow velocity measurements of the five-hole probe to 5 determine the flow velocity in cartesian coordinates u, v, w, and subtracting the contribution to the relative wind speed caused by the rotation of the blade (Petersen et al., 2015). The axial wind vector component, u, is affected by the axial induction of the rotor, which can be calculated using an equation derived by Madsen et al. (2010):

$$a = 0.0892C_T^3 + 0.0545C_T^2 + 0.2512C_T \tag{4}$$

where  $C_T$  is the thrust coefficient derived from the loading on the actuator disc, expressed via the lift and drag coefficients 10 projected to the axial direction,  $C_y$  (Madsen et al., 2010):

$$C_y = C_l \sin(\phi) - C_d \cos(\phi) \tag{5}$$

The infinitesmal thrust dT on an annular element of length dr can then be defined as (Madsen et al., 2010):

$$dT = \frac{1}{2}\rho V_{rel_{xy}}^2 C_y c N_B dr \tag{6}$$

where  $V_{rel_{xy}}$  is the relative velocity projected onto a section of the blade,  $N_B$  is the number of blades and  $2\pi r$  is the swept 15 distance of the blade element. The local thrust coefficient is then defined as:

$$C_T = \frac{dT}{\frac{1}{2}\rho U_{\infty}^2 2\pi r dr} = \frac{V_{rel_{xy}}^2 C_y c N_B}{U_{\infty}^2 2\pi r}$$
(7)

By assuming an initial  $U_{\infty}$  value and iterating through equations (7), (4) and (3), the upstream wind profile can be calculated. Petersen et al. (2015) demonstrated the method using five-hole probe measurements taken from the DANAERO project (Madsen et al., 2010), as well as simulations. The method proposed was shown to provide accurate estimations of the flow field upstream when compared with independent measurements and simulations.

# **3** Theoretical Models

While experimental measurements of the wind turbine blade angle of attack are lacking in the literature, several theoretical models for the variation in  $\alpha$  with azimuthal position and yaw angle have been developed. The simplest model for calculating

the angle of attack can be derived from the basic velocity relationships for an airfoil, where the angle of attack,  $\alpha$ , is defined as the angle between the rotational velocity of the turbine rotor and the relative wind velocity vector, W. This geometric angle of attack can be determined using equation (8) (Morote, 2015):

$$\alpha_{geom0} = \arctan \frac{\cos \beta - \lambda_r \sin \beta}{\lambda_r \cos \beta + \sin \beta} \tag{8}$$

where  $\alpha_{geom0}$  is the geometric angle of attack in axial flow conditions,  $\beta$  is the blade pitch at the radial location of interest, *r*, and  $\lambda_r$  is the local speed ratio at the same location.

While equation (8) is appropriate for a quick estimation of the blade angle of attack, the error associated with neglecting the axial and tangential flow induction factors (a and a', respectively) associated with the turbine in a three-dimensional space can be detrimental to the accuracy of the prediction. When the wind turbine is yawed, determining the angle of attack becomes more complex as it varies with the azimuthal position of the blade,  $\phi$ . Burton et al. (2011) present a more complete model for

calculating the angle of attack by combining several models and correction factors to accomodate these variations.

While the model reported by Burton et al. (2011) has historically been considered sufficient, a more recent publication by Morote (2015) presents a new model for calculating the effective angle of attack,  $\alpha_{eff}$ , based again on the geometric angle of attack defined in equation (8). In this model, interference functions are defined to accommodate the axial interference and span-wise interference, labeled as f(r) and g(r) respectively. From Morote (2015),

$$f(r) \approx 1 - \frac{a_0 \lambda_r}{(1 + \lambda_r^2) \tan(\alpha_{geom0})} \tag{9}$$

where  $a_0$  is the radially dependent induction factor for axial flow at the blade lifting line, and

$$g(r) \approx -\sqrt{A^2 + B^2} \tag{10}$$

where

$$A = \frac{a_{a0}}{(1+\lambda_r^2)^2 \tan(\alpha_{geom0})} \left(\frac{\lambda_r (1+\lambda_r^2)(r/R)}{2(1+a_{a0})}\right)$$
(11)

$$B = \frac{a_{a0}}{(1+\lambda_r^2)^2 \tan(\alpha_{geom0})} \left(\lambda_r^2 - 1 - \frac{2\lambda_r}{\sin(2\alpha_{geom0})}\right)$$
(12)

Here,  $a_{a0}$  is the radially dependent azimuthally averaged induction factor for axial flow. The equation for g(r) is only valid at certain radial positions, which means a phase shift  $\Delta$  is also required. This phase shift accounts for the change in azimuthal

position of the blade as an air particle travels over the blade chord, which would result in a slight variation in the angle of attack along the chord line. The phase shift can be calculated as:

$$\Delta \approx \arctan(\frac{B}{A}) \tag{13}$$

Combining the interference functions and phase delay with the geometric model identified in equation (8), the Morote (2015) 5 model for calculating the effective angle of attack is defined as:

$$\sin \alpha_{eff} = \sin \alpha_{geom} [f(r) + g(r)\delta \sin(\phi + \Delta)]$$
(14)

where  $\delta$  is  $\sin \gamma$ .

While Morote (2015) has compared modeled pressure distributions over the blade to experimental data, neither Burton et al.
(2011) or Morote (2015) present any experimental measurements of the angle of attack to validate their mathematical models. In fact, other than the studies discussed previously, there is a significant lack of experimental validation data concerning the variation in the angle of attack with turbine yaw and azimuthal position.

### 4 Development and Calibration of a Five-Hole Pressure Probe

The custom five-hole pressure probe and in-blade data acquisition system (DAQ) used here was constructed and calibrated 15 in-house and described in Moscardi and Johnson (2016). The probe was calibrated in an open jet wind tunnel for pitch and yaw positions of  $\pm 50^{\circ}$ . Calibration experiments were conducted at the blade design Reynolds number with several methods evaluated to determine pressure coefficients.

Moscardi and Johnson (2016) used cubic interpolation, as well as equation (1) to calculate  $\alpha$  from pressure distribution measurements. More DAQ and blade details are provided in the following section.

# 20 5 Experimental Methods

All experiments were conducted at the University of Waterloo Wind Generation Research Facility, which houses a fan bank capable of generating wind speeds up to 13 m/s in a facility 15.4 m wide, 19.5 m long, 7.8 m high at the sides. Flow fields generated by the facility have turbulence intensities in the range of 5.9% to 6.2%, and the blockage ratio of the test turbine in the facility is approximately 7%. The combination of high turbulence and low blockage is considered ideal for wind turbine

#### 25

testing, as it is representative of the environmental conditions that wind turbines would be exposed to in the field. More detailed information about the facility geometry and generated flow field can be found in Moscardi and Johnson (2016).

The test turbine is a custom-built 3.4 m diameter upwind horizontal-axis wind turbine, pictured in Figure 3. The rotor for these experiments consisted of one 3D-printed blade formed from five identical modules and two dynamically balanced rods.