# Peer review of "Flow angle measurement of a yawed turbine and comparison to models"

_Wind Energy Science, 2016_

## Referee Comment (RC1) · Anonymous Referee #1 · 7 Mar 2017

General comments:

The paper presents the results concerning the measurement of the angle of attacks of a yawed wind turbine model tested within a wind tunnel.

I personally do not like that the introduction session does not contain the state-of-the-art review, which seems anyway modest, not complete and focused on the Petersen et al. (2015) research, which concerns with the estimation of the far field velocity by using Multi-hole probes rather than with the measurements of the angle of attack for a yawed wind turbine. Moreovoer, it is not clearly highlighted the novelty of the approach presented in this paper.

The papers aims also at comparing the experimental data with the predictions of the model developed by Morote (2015). The readers must therefore be capable of understanding, by reading the sole paper, the theoretical basis of the model, so as it must be clear the sequence of equations used by the model, which are the model inputs and how these are gathered (measured, modeled?) The description of the Morote model is however poor and unclear (see further detailed comments), and forces the readers to read the Morote paper just to gather a basic understanding of how the model works.

Moreover, it is claimed that the Morote model is more accurate if compared to the predictions of the model presented in Burton (2011). Considering this last sentence, as well as the paper title "Flow angle measurement of a yawed turbine and comparison to models", the experimental data must be compared with the predictions of more than one model. For example, data should be compared to the Burton model prediction, so as to prove that the Morote one is better. The paper must therefore provide enough information concerning the Burton model and must be highlighted the differences among the Burton and Morote models.

To support this last statement, consider the results shown in Figure 10 and 11, which clearly highlight that predictions are getting worse as higher the axial induction factor is (at r/R = 0.72 the axial induction factor is higher than r/R = 0.55). I expect that the predictions will be even worse for axial induction factor typical of multi-MW 3-bladed wind turbines. It is clear that the goal of this paper is not the one of presenting and validating a new method (Morote), but rather to compare exp. data with numerical predictions obtained using different models. However, if only the comparison between exp. data and Morote predictions is given, it seems that the goal of the paper is to validate a model which clearly seems not appropriate to predict the angle of attacks of either an-yawed wind turbines characterized by typical values of the axial induction factor. The only way to have this paper published in this journal is therefore to include comparisons between the exp. data and other numerical predictions.

Detailed comments

Page1, Line 24-25: "The angle of attack is therefore calculated using models (e.g. the

blade-element momentum method) using measurements several steps removed from the leading edge of the blade". Unclear sentence, please rephrase it. . Page 2, Line 20: The local flow angle (LFA) is indicated as \alpha in figure 1. Please change the figure

Page2, Line 28-29: "was essentially constant although they found a variation in \alpha of as much as 2.5 with azimuthal position that they described as the influence of the atmospheric boundary layer". Are the authors referring to the angle of attack or the local flow angle? Moreover, please make use of the punctuation so as to have a better readable sentence.

Page 3. Line 13-15. "These values were then used to calculate the blade angle of attack. Johnson et al. (2012) found that the method resulted in angle of attack estimates within 15-20% of airfoil design data indicating that the use of velocity measurements to validate five-hole probe measurements may be appropriate". Unclear sentence. What exactly means "within 15-20% of airfoil design data"? do you refer to the angle of attack at which the airfoil should operate at rotor optimal operating conditions?

Page 4. Line 5-6. "and subtracting the contribution to the relative wind speed caused by the rotation of the blade". Should it be: subtracting the contribution of the relative wind speed caused by the...?

Page 5. Line 5. "$\beta$ is the blade pitch at the radial location of interest" It should be clarified that $\beta$ is the sum of the twist at the radial location of interest and the blade pitch, which is instead radially constant

Page 5. Line 17. "where a0 is the radially dependent induction factor for axial flow at the blade lifting line" Page 5. Line 23 "Here, aa0 is the radially dependent azimuthally averaged induction factor for axial flow" What exactly is the difference between these two parameters? How are they gathered?

Page 5. Line 25. "The equation for g(r) is only valid at certain radial positions, which

means a phase shift $\Delta$ is also required. This phase shift accounts for the change in azimuthal position of the blade as an air particle travels over the blade chord, which would result in a slight variation in the angle of attack along the chord line." g(r) and f(r) are radial-dependent function. What does it mean that they are only valid at certain radial positions? The text gives the idea that the phase shift is necessary to account for the change in azimuthal position that results in a variation of the angle of attack along the chord line. Does this mean that g(r) and f(r) are only valid at certain azimuthal positions, and therefore must be corrected of the phase shift?

Page 6. Line 6 Please replace $\delta$ with $\sin\gamma$ in the equation 14. There is no need of introducing extra symbol

Page 6. Line 6 The $\alpha$geom used in equation 14 is computed by means of Eq. 8 even when when the flow is not aligned with the rotor disk, i.e. when $\gamma$ is different from 0? Please be more specific.

Page 8. Line 6-7. "The scale of these values was confirmed by the five-hole probe measurements using a method described by Petersen et al. (2015)." Please include the values computed with this method.

Page 8. Line 8. "Table 2. Summary of Model Axial Induction Factors." The Morote model makes use of a0(r) and aa0(r). Plese add a figures that report these values wrt the radial distance r. Explain also how they are calculated with PROPID and if, and how, they were compared with values gathered with the Petersen et al. method.

Page 8. Line 12-13. "Axial induction factors calculated using PROPID (PROPID, 2016) were used in combination with the Biot-Savart law to adjust local flow angle measurements to the blade angle of attack." Eq. 1 is used to compute the the local flow angle, and the introduction makes reference to Schepers and van Rooij (2008) and Shen et al. (2009) to explain how the angle of attack can be derived from the local flow angle by correcting for induction due to the bound circulation. The authors, however, claim that they are using a different approach. Please provide more details about the adopted

approach and how it differs from the ones of Schepers and van Rooij (2008) and Shen et al. (2009)

Page 9. Line 1. "slight non-uniformity in the flow at the Wind Generation Research Facility, as described by Best (2010)" Please depict the non-uniformity of the flow in a figure, and highlight that it could lead to not-negligible (3-4 degrees) variations of the angle of attacks like those reported in figure 5. This could be done, for example, by using the prediction depicted in Figure 12 to compute an azimuthally-variable $\alpha$geom, that, in turn, can be used to compute $\alpha$eff with eq. 14.

Page 11 Line 1-2. "Discrepancies at $\lambda$ = 3.6 are likely caused by inaccuracies in the calculated axial induction factors, as was confirmed by modeling the $\alpha$ distribution with arbitrarily changed a value" It would be usefull to see the dependency of the angle of attack to the induction factor, at least for one tsr.
* * *

---

## Referee Comment (RC2) · Anonymous Referee #2 · 13 Mar 2017

The manuscript addresses the measurements of the flow angles of a rotating blade subject to normal and to yawed flow.

As usual the authors start with an overview of the published literature. Unfortunately this overview is biased, and not covering all important activities in the field. As an example all the contributions and contributors to the IEA wind task 29, where a.o. yaw modelling and yawed flow experiment are extensively treated, are absent. But more serious is the set-up of the experiment and the results presented in the manuscript.

Instead of a full configuration, there is only one of the three blades physically present. The other two blades are replaced by bars, intended for dynamically balancing the rotor. The aerodynamics are thus not representative for a wind turbine rotor. It is just one single blade, operating in an environment, polluted with the viscous wake and with

the added forces of two rotating rods (with a lot of viscous drag).

But the most serious issue can be seen when observing the figures 5 and 10. These figures show the measured AoA (derived from the measured angles with the 5 hole pitot tube in front of the blade) as a function of azimuth angle. Since this is a kind of wind tunnel set-up, the measured AoA should be constant over the azimuthal position of the blade. But it is not, as can be seen in figures 5 and 10.

Depending upon the tip speed ratio lambda and the radial position (fig 5 r/R=0.55 fig 10: r/R=0,72) the "measured" variation in AoA over the azimuthal position is typically +/- one to two degrees!!

The authors do not provide any explanation for this variation. They only refer to earlier work of Maeda and Kawabuchi (2005). But the experiments of these authors refer to measurements in an outdoor environment and there variations of AoA over azimuthal position can be expected due to the influence of the atmospheric boundary layer. Here such ABL simulation is absent, that is at least the presence of an ABL simulation on the facility it is not mentioned, nor quantified by the authors. So the reviewer has to assume the rotor operates in a uniform inflow (with a turbulence level of about 6%)

And this has as consequence that nor fig 5 neither fig 10 can be properly understood by the reader. Why this periodic behaviour of the AoA?

And with an inaccuracy of up to +/- 2 degrees ( the difference between the expected constant AoA over azimuthal position and the measured AoA's) the interpretation and the comparison with numerical codes and/or empirical models for yawed flow also become useless. On other words, to my opinion the present manuscript (lacking information and explanation and hence raising a lot of unaddressed, let alone answered, questions) has not a single added value to the scientific wind energy community.

---

## Author Comment (AC1) · 29 May 2017

Thank you for the careful and detailed review of our experimental study.

The general subject of a wind turbine in yaw is significant and a state-of-the-art review would encompass more than what would be required to introduce these experimental measurements. The literature that has been reviewed is specific and of recent publication. Updates to the literature review are always possible if they address the specifics of the experiments detailed here.

Space was dedicated to the description of the Petersen et al. paper to properly explain the method used later in the paper to assess the upstream wind field using angle of attack measurements. It was felt that this was an exciting addition to the paper, as it both demonstrates the potential of this method as well as helps to explain and quantify

variations in the upstream flow stream, as evident in the angle of attack vs. azimuthal position figures.

The authors agree that the description of the Morote model is brief and was limited due to space constraints in the paper. However, if this limited the clarity of the explanation, the background given can be expanded to better explain the model.

It is actually not stated that the Morote model is more accurate, only that it is more recent. Due to the recent publication, the author's decided to focus the paper on this one comparison. The angle of attack was also completely modeled using the Burton et al. method and compared to experimental data, but this was left out due to space limitations.

The paper could be edited to add a background review of the Burton et al. methodology and the resulting model predictions. The model results are in close agreement with each other throughout the test conditions considered. This is an experimental study with some modeling to show the results.

In the review comments, there is considerable discussion of the model results and modeling in general. The main focus of the experimental submission (Title: Flow angle measurement...) was to detail the experimental methods (onboard instrumentation, 3D printed blade etc) and uniqueness of the facility.

Specific comments

Page 2, Line 28-29 Authors Maeda and Kawabuchi refer to the angle alpha as the "angle-of-attack," though no acknowledgement was given in the publication of a conversion from LFA to angle of attack (can not comment if it was done).

Page 3, Line 13-15 This statement can be revised for clarity. Previous research has shown that reasonable results for the airfoil performance (essentially 2D airfoil data) can be obtained from fully 3D rotating wind turbine blades using multi-hole probes.

Page 4, Line 5-6 This sentence can be phrased more clearly. It is referring to the

subtraction of the tangential velocity vector caused by the rotation of the wind from the relative wind velocity in the velocity triangle. In other words, subtracting omega*r from W to get Uinf.

Page 5, Line 5 This can be reworded or clarified before the final edit.

Page 5, Line 17 The difference between the parameters is that a0 is the induction factor for axial flow, and can vary throughout the rotation with azimuthal position. In contrast, aa0 is averaged throughout all azimuthal positions. The induction factors were calculated using the PROPID input code that was used to design the wind turbine blade.

Page 5, Line 25 Yes, that is exactly correct. According to the Morote paper, the interference functions are only valid at specific azimuthal positions and must be corrected using the phase shift. This explanation can be clarified.

Page 6, Line 6 Good point. This symbol was only introduced to maintain consistency with the Morote paper, in which it is also introduced.

Page 6, Line 6 Yes, according to the Morote model, Equation 8 is always used to calculate the geometric angle of attack. The induction correction factors account for the variation in yaw angle. This can be clarified before the final edit.

Page 8, Line 6-7 These values can be added to the final version of the paper if space allows.

Page 8, Line 8 The induction values calculated in using PROPID and presented in Table 2 were used as a0 and aa0 inputs to the Morote model. The assumption of a0 = aa0 was considered appropriate given the expected uniform flow field in the small-scale measurements. The values were calculated using PROPID using the input code originally used to design the blades. The calculation itself was done iteratively following the BEM method. These points can be clarified in the final edit.

Page 8, Line 12-13 The paper doesn't state that we used a different approach than

Schepers and van Rooij (2008) or Shen et al. (2009). We followed their approach and similarly used the Biot Savart law to convert from the LFA to angle of attack.

Page 9, Line 1 The non-uniformity is depicted in Figure 12 on Page 15, in which the Petersen method has been used to calculate the upstream flow velocity as a function of azimuthal position. A reference to this plot can be made earlier to highlight the influence of the non-uniformity on the flow field.

Page 11, Line 1-2 This can be added in an additional figure if space allows.

---

## Author Comment (AC2) · 29 May 2017

Thank you for the careful and detailed review of our experimental study.

The general subject of a wind turbine in yaw is significant and a state-of-the-art review would encompass more than what would be required to introduce these experimental measurements. The literature that has been reviewed is specific and of recent publication. Updates to the literature review are always possible if they address the specifics of the experiments detailed here. The authors are well-aware of IEA Task 29. IEA Task 29 is important although full results of the yawed cases are somewhat challenging to find in the peer reviewed literature. If angle of attack measurements in that Task have been missed by the authors they will certainly be welcomed and added to the background.

We have many potential rotor configurations (1-3 blades) and have undertaken many

studies including single blades but more commonly with a conventional 3 bladed rotor when that was the focus. Since the blade was 3D printed and contained all the instrumentation in the one blade the remaining rods are used for balancing. This approach was considered acceptable given the magnitude of the testing wind velocity. The wake following the very slender weighted rods did not alter the measured results in side by side comparison. However, the single-bladed rotor does result in a significantly lower induction than a typically 3-bladed rotor. This is reflected in the low induction a values presented in Table 2.

The design of the wind facility is not that of a conventional wind tunnel. Turbulence intensity is intentionally high to attempt to replicate our field experience measurements. It is stated in the paper that the wind generation facility experiences a non-uniformity in the flow from the current fan configuration, as described by Best (2010). The non-uniformity was quantified by calculating the upstream flow velocity as a function of the azimuthal position using the Petersen (2015) method and measured five-hole probe data (see Fig. 12). This non-uniform flow field is reflected in the angle-of-attack as a cyclical variation. However, when the variation is accounted for, the theoretical and experimental values are considered to be in close agreement.

The periodic behaviour in the AoA during axial conditions is due to the consistent non-uniformity of the flow field. It is not related to the presence of an atmospheric boundary layer, as stated by the reviewer.

The uniqueness of the experimental design (3D printed blade with self-contained data acquisition) and the application in the wind facility do provide substantial contributions to the wind energy research community.
* * *